# Evaluation of Control Strategies for *Xylella fastidiosa* in the Balearic Islands

**DOI:** 10.3390/microorganisms10122393

**Published:** 2022-12-02

**Authors:** Bàrbara Quetglas, Diego Olmo, Alicia Nieto, David Borràs, Francesc Adrover, Ana Pedrosa, Marina Montesinos, Juan de Dios García, Marta López, Andreu Juan, Eduardo Moralejo

**Affiliations:** 1TRAGSA Group, 07009 Palma de Mallorca, Spain; 2Institute of Agro-Food and Fisheries Research and Training of the Balearic Islands (IRFAP), 07009 Palma de Mallorca, Spain; 3Agriculture Service, Ministry of Agriculture, Fisheries and Food of the Balearic Government, 07006 Palma de Mallorca, Spain

**Keywords:** risk assessment, phytosanitary measures, invasive pathogens, vector-borne disease, *Philaenus spumarius*

## Abstract

The emergence of *Xylella fastidiosa* (Xf) in the Balearic Islands in October 2016 was a major phytosanitary challenge with international implications. Immediately after its detection, eradication and containment measures included in Decision 2015/789 were implemented. Surveys intensified during 2017, which soon revealed that the pathogen was widely distributed on the islands and eradication measures were no longer feasible. In this review, we analyzed the control measures carried out by the Balearic Government in compliance with European legislation, as well as the implementation of its control action plan. At the same time, we contrasted them with the results of scientific research accumulated since 2017 on the epidemiological situation. The case of Xf in the Balearic Islands is paradigmatic since it concentrates on a small territory with one of the widest genetic diversities of Xf affecting crops and forest ecosystems. We also outline the difficulties of anticipating unexpected epidemiological situations in the legislation on harmful exotic organisms on which little biological information is available. Because Xf has become naturalized in the islands, coexistence alternatives based on scientific knowledge are proposed to reorient control strategies towards the main goal of minimizing damage to crops and the landscape.

## 1. Introduction

No other activity of humanity has previously been impacted the negative side effects of globalization like agriculture and forestry [1]. Since the beginning of the Modern Age, the economic damage caused by invasive pathogens and pests to crops and forests has had a considerable impact [2,3]. Even today, between 20% and 40% of harvests in main crops are lost directly or indirectly due to pathogens and pests [4]. To reduce the risk of introducing alien pathogens while minimally interfering with international trade, phytosanitary regulations have been accorded among countries through the International Plant Protection Convention (IPPC). In Europe, although plant health legislation is implemented with high standards and scientific consensus [3], it has failed to prevent the entry of harmful alien pathogens, such as *Phytophthora ramorum, Hymenoscyphus fraxineus*, and *Xylella fastidiosa* [5,6,7].

The bacterium *Xylella fastidiosa* (Xf) native to the American continent is one of the most feared phytopathogens in the world due to the damage it causes to crops of great economic value such as grapevine, citrus, coffee, almond, and olive trees, among others [8]. Xf is transmitted non-specifically by insects belonging to sharpshooter leafhoppers (Hemiptera: Cicadellidae, Cicadellinae) and spittlebugs (Hemiptera: Cercopoidae) [9]. Once introduced into the xylem vessels, the bacterium colonizes the vascular system of the plant, compromising the water supply in susceptible hosts. Xf’s bad reputation stems mainly from the fact there are no known curative treatments for infected plants [10]. Although experimental injection of antibiotics (e.g., streptomycin) into infected plants reduces disease symptoms, the infection is not eliminated [11,12] and reappears if treatment is stopped. Furthermore, antibiotics are banned in the EU as a treatment against bacterial plant diseases [13]. Several minerals and compounds in addition to microbial endophytes as biocontrol agents have been tested in America and Italy, showing some protection against Xf infections in grapevines [14], citrus [15], and olive trees [16]; however, their large-scale application is still economically expensive [13]. For this reason, efforts are also being made to find alternative control systems to those directed solely at the bacterium as an economically sustainable strategy over time.

Other factors adding difficulties to control strategies are the diversity and wide host range of the pathogen [17]. Three main subspecies with allopatric origins, *pauca*, *fastidiosa* and *multiplex*, are known from South, Central, and North America, respectively [8]. Within each subspecies, diverse genetic lineages with different host ranges have evolved [14]. Genetic recombination among subspecies seems to provide the main source of genetic variation, which may lead to host jumps [17,18]. To date, Xf as a taxonomic unit is known to infect over 638 plant species [19]. However, each of the 90 known sequence-type (ST) profiles has smaller host ranges, sometimes with overlapping hosts [20,21].

Although there have been some previous unconfirmed reports [22], Xf was first detected in Europe on olive trees in Apulia, Italy [7]. The olive quick decline syndrome (OQDS) induced by Xf has destroyed millions of olive trees in Apulia since 2013, causing a significant economic and landscape impact [23]. In 2015, Xf was detected on the island of Corsica and shortly after in the Provence-Alpes-Côte d’Azur in France [24], followed by the Balearic Islands in October 2016 [25]. Currently, the pathogen is also established in Alicante (Spain) [26] and Israel [27], and new outbreaks have emerged in Tuscany [28] and Portugal [29]. The origin of these introductions and their recent evolutionary history have been investigated [30]. This is a necessary first step to search for control strategies, as will be explained later.

In the Balearic Islands, several factors have converged to considerably delay Xf detection: (i) Xf strains are not excessively virulent on the main hosts, almond and wild olive trees; (ii) symptom development coincides with the summer drought peak; (iii) the presence of other wood or root pathogens is already established causing basal mortality rates enhanced by drought; (iv) there is a lack of renewal of an ageing almond tree population together the abandonment of land care due to low yields; and finally, (v) and perhaps most importantly, Xf was not expected.

Since Xf first detection in the Balearic Islands in 2016, phytosanitary measures, included in Decision (EU) 2015/789 and the updated Regulation (EU) 2020/1201, have been implemented, while additional measures have been enacted by regional and Spanish authorities. At the same time, research programs were initiated to find out the chronology of the introductions and the incidence of the pathogen in crops. To date, Xf in the islands shows one of the largest genetic diversities in Europe, while their impact on various crops and forest species allows for an evaluation of the implementation of quarantine measures in a varied context. We believe that the pathogen’s situation in the Balearic Islands is illustrative of the difficulties of legislating on introduced organisms with a complex and poorly understood biology and wide host range. Paradoxically, Xf would likely have entered Mallorca in 1993, around 22 years before the legislation to be applied to control the spread of the pathogen in Europe was launched and eight years before being in force the Council Directive 2000/29/EC on protective measures against the introduction and spread of organisms harmful to plants inEurope. In this review, we expose the chronology of the events and how, as the research on the origin and phylogenetic relationships of the Xf populations of the Balearic Islands progresses, the perception of the effectiveness of phytosanitary measures is changing. Our purpose is to focus exclusively to control strategies in the Balearic Islands, as excellent general reviews of control attempts and management of Xf have been recently published [31]

## 2. Action Plans to Combat *Xylella fastidiosa* in the Balearic Islands

On November 25, 2016, an outbreak of *Xylella fastidiosa* (Wells et al. [32]) was officially declared in the Balearic Islands, and thereby a containment plan was adopted to eradicate and control it after the Resolution of the Ministry of the Environment, Agriculture and Fisheries of the Balearic Islands Government. A demarcated area was established around the outbreak focus in a garden center in Porto Cristo, Mallorca, and an infected and buffer zone was defined (Article 4, Decision 2015/789). The action plan for the implementation of phytosanitary measures was coordinated by the phytosanitary authority, the General Directorate of the Agriculture Department of the of Agriculture, Livestock and Rural Development of the Government of the Balearic Islands in collaboration with the other local administrations and the General Directorate of Natural Spaces and Biodiversity. In addition, a management and coordination group together with a scientific group were designated by the official body. A mixed commission between the Balearic Government and the Civil Guard was formed to coordinate the controls at airports, ports, and roads and prevent the spread of the Xf outside the Balearic territory and between islands.

Mandatory surveys began with the entry into force of the decision and were intensified following the first detection of Xf. The Official Plant Health Laboratory of the Balearic Islands (hereinafter LOSVIB) was in charge of analyzing plant samples collected in the demarcated zone and beyond at the end of 2016. The result analysis successively revealed that the pathogen was widely established in Mallorca, Ibiza, and Menorca. Consequently, the buffer zone had to be reviewed at each new outbreak communication, soon extending to almost the entire surface of the islands in June 2017 (Figure 1). At the same time, the local media echoed the situation of the new pathogen, causing some concern in public opinion [33].

From October 2016 to the end of 2022, significant resources have been dedicated to understanding the situation of Xf in the field. The LOSVIB has analyzed more than 18,016 plant samples collected from a total of 382 plant species. In the 32 notifications to the SG SANTE, 1360 Xf-qPCR positives and 37 species as host plants have been declared (Figure 2, Table 1). All samples were tested according to the EPPO Diagnostic Protocol (PM 7/24(4), [29]) both on symptomatic and asymptomatic plants, as well as on insect vectors. All new qPCR-Xf-positive hosts were confirmed by the National Reference Laboratory for phytopathogenic bacteria in Valencia, Spain, and their genetic profiles were determined by MLST in the IAS-CSIC laboratory in Córdoba, Spain. To date, three subspecies and four sequence types (ST1, ST7, ST80, and ST 81) have been identified on the islands [30], and the genomes of several isolates have been sequenced [34,35]. More details on the situation of Xf in the Balearic Islands can be found in Olmo et al. [36].

As surveys intensified, awareness grew that the strict application of Decision 2015/789 was not possible in the Balearic Islands. These concerns of the official body were discussed and exposed in the commission audits of June 2017. Shortly afterwards, as a result, the Balearic Islands were declared an infected area in the Execution Decision (EU) 2017/2352 of the commission of 14 December 2017, which amends Execution Decision (EU) 2015/789. Since then, containment measures could be applied instead of eradication measures in Corsica and the Balearic Islands. In both territories, the bacterium was widely established and could no longer be eradicated. Elimination was limited to infected and positive plants within a 50 m radius. In the new Regulation (EU) 2020/1201 in force that repeals Decision 2015/789, these areas are included in Annex III, with the containment measures focused on minimizing the amount of bacterial inoculum and keeping the vector population in check, the lowest possible levels. The containment measures are applied in the infected areas and are aimed at the elimination and destruction exclusively of those infected plants detected within the framework of the annual control that is carried out in certain places within the vicinity of plant complexes with cultural and social interest.

### 2.1. Elimination of Plants in Infected Areas

All plant samples that enter the LOSVIB are registered in an official database that includes relevant information on the host, location, date, etc., as well as the coordinates where the sample was collected. Decision (EU) 2015/789 required all member states to eliminate all host plants, regardless of their phytosanitary status, within a 100 m radius around the Xf-positive plant. In the amendment of the decision in December 2017, given that the strict application of Article 6 would mean wiping out the entire agricultural landscape, the phytosanitary status of the Balearic Islands was modified by declaring the entire archipelago as an infected zone. In the new status, the containment plan approved allows only the affected plants to be removed, ensuring that no new plant infections occur within a radius of 50 m.

In the currently in-force Regulation (EU) 2020/1201, all qPCR-positive plants detected in an infected zone are eliminated, except plants for scientific purposes or with particular cultural or social value. Since October 2016, a total of 13,757 plants were eliminated on all the islands, 10,113 plants in Mallorca, 329 in Menorca, 3314 in Ibiza, and 1 in Formentera. Of these eliminations, only 1276 were positive samples for Xf, with a total of 37 host species in all the Balearic Islands (Table 1). The distribution by islands is as follows: 773 positive samples were from Mallorca, 369 from Ibiza, and 218 from Menorca, obtaining together a total of 1360 positive samples, and thus approximately 94% of the positives were eliminated.

### 2.2. Measures against Vectors of the Specified Pest in Containment Zones

In compliance with Article 7 of Decision (EU) 2015/789, appropriate agricultural practices for vector management in the demarcated area have been promoted by the official body. Training courses have been held for farmers on the management of vectors in crops. In 2018, a phytosanitary supply campaign was started to combat Xf. Among the containment measures, farmers have been subsided for the use of the following authorized insecticides: Kaolin 95% WP p/p; Deltamethrin 10% (EC) p/v; and Lambda-Cyhalothrin 10% (CS) p/v on olive trees; Deltamethrin 10% (EC) p/v, Lambda-Cyhalothrin 10% (CS), Pyrethrin 4% (EC) p/v, and Azadirachtin 3,2% (EC) p/v on grapevines; and Deltamethrin 10% (EC) p/v and Lambda-Cyhalothrin 10% (CS) on almond trees. Between 2019 and 2021, a total of 2675 ha of almond plantations, 1888 ha of olive trees, and 1440 ha of vineyards have been treated with insecticides to control vector populations.

### 2.3. Annual Surveillance of Infected Areas

Until the amendment of Decision (EU) 2015/789, with the entry into force of Regulation (EU) 2020/1201, censuses and two annual samplings of host plants and vectors were carried out within a 100 m radius of the positive sample point, including one during the vector’s flight season. Since 2018, sampling has been reduced to an annual one with a 50 m radius. In both cases, Xf-positive plants within the radius were eliminated. Until 2020, 1563 samples were analyzed. Since 2020, the risk-based estimate of the system sensitivity tool RIBESS+ has been applied to sample the specified plants in places that have particular cultural, social, or scientific values.

A total of 3001 samples were recorded during the 2018–2021 period that were within the infected zone, and of these, 76% of the infected radios were inspected. Given that the average density of hosts is 100 for every 50 m of radius and the occupied surfaces are more than 673 ha, the measure supposes analyzing the unfeasible quantity of more than 60,000 samples per year. Within the areas of interest with social and cultural value, careful monitoring of the germplasm banks has been carried out on 499 crop varieties in four fields in Mallorca, 117 varieties in Ibiza in one field, and 448 in Menorca in one field.

### 2.4. Authorization Regarding the Planting of Specified Plants in Infected Areas

A strict application of Article 5 of the EU Decision 2015/789 that prohibits the planting of hosts in infected areas would have condemned almond tree plantations and vineyards in Mallorca and olive trees in Ibiza to extinction. Since 2017, massive sampling and field studies have been carried out in which information has been collected on the epidemiological situation of different crop varieties. At the same time, inoculation tests have been conducted in an insect-proof net tunnel exposed to environmental temperature to determine the most resistant or susceptible varieties. In addition, germplasm banks of crop varieties have been intensively monitored and sampled during the past years. This information has been used to legislate the planting authorization of certain varieties through the Resolution of 14 February 2018, of the General Directorate of Agricultural Production Health, which approves the request of the Autonomous Community of the Balearic Islands to the planting of certain host plants of Xf in infected areas.

New almond plantations have been allowed, except for the following list of varieties: Marcona, Garrigues, Bord de Santa Maria, Bord de Selva, Bord des Raiguer, Corona, Filau, Lluca, Menut, Mollar, Morro de vaca, Pere Gelabert, Pintadeta, Rutlo, Trinxets, Desmai Victoria, Viveta, and Vivot. For olive tree (*Olea europaea* var. *europaea*), the following varieties are allowed: Empeltre, Mallorquina, Arbequina, Picual, Arbosana, Koroneiki, Hojiblanca, Cornachuela, Cornicabra, Morruda, Sikitita, and Frantoio. For winemaking (*Vitis vinifera* L.), the varieties authorized are Cabernet Sauvignon, Callet, Chardonnay, Escursac, Fogoneu, Garnacha Blanca, Garnacha Negra, Giró Ros, Gorgollasa, Macabeo/viura, Malvasía aromatica/Malvasía de Banyalbufar, Manto Negro, Merlot, Moll/Prensal Blanc/Prensal, Monastrell, Muscat of Alexandria, Muscat Petit Grans, Parellada, Petit Verdot, Pinot noir, Riesling, Sauvignon Blanc, Syrah, Tempranillo, and Viognier.

In addition, the launch of New Generation funds is promoting a restructuring plan for rainfed fruit plantations in the Balearic Islands for the period 2021–2027. The objectives of the plan are the recovery of part of the lost area, the improvement of the efficiency in the production and transformation, and the increase in the commercialization of derived or elaborated products raising the agricultural income associated with the agriculture sector.

### 2.5. Movement of Plant Material out of the Balearic Islands and Between Islands, Phytosanitary Passports, and Border Controls

Articles 4 and 9 of Decision 2015/789 banned the export of plants from demarcated areas. These measures were reinforced with the Ministerial Order APM/21/2017 of specific prevention measures; the Resolution of the Minister of the Environment, Agriculture and Fisheries, which prohibits the departure from the territory of the island of Ibiza to the rest of the Balearic Islands; and the declaration of public utility in the fight against Xf through decree 65/2019. It prohibits the exit from the territory of the Balearic Islands for all plants for planting, except seeds, belonging to the genera or species listed in Annex I of the Execution Decision (EU) 2015/789, which is maintained by Regulation (EU) 2020/1201.

Fortunately, the Mediterranean Sea provides an effective natural barrier to the spread of Xf outside the islands, as indicated in the absence of the pathogen on the island of Formentera, only a 3 km distance from Ibiza [36]. Given the importance of the movement of tourists in the Balearic Islands, priority has been given to controlling the departure of plants from airports and ports. The exit of unauthorized plants is prohibited, as well as circulation between islands. Controls are carried out by the state security forces and authorized customs inspection agents in ports and airports in collaboration with the Plant Health Service of the Balearic Islands (Decree 65/2019). To support the control measures, informative posters and controlled containers for passengers have been placed at the airport control points (Figure 3). Since 2017, the Civil Guard has carried out more than 65,432 passenger inspections, and 417 inspections have been carried out in ports and airports in customs areas, registering 129 incidents with the interception of infected plant material.

Plant production for export outside the islands is negligible. Between 2017 and 2022, the official body only received one request for authorization to export Xf host plants outside of the Balearic Islands. In 2020, a bonsai production nursery was authorized after verifying compliance with articles 19 and 20 of regulation 2020/1201. In total, 204 authorized plants, mainly *Olea europaea* var. *sylvestris* and *Myrtus communis*, have been exported to mainland Spain and other European countries.

Most ornamental plants that are sold in small retailers and garden centers (plant operators) in the Balearic Islands are imported. Inspections of plant phytosanitary passports have been conducted on the 143 registered professional plant operators. More than 180 health inspections have been carried out in nurseries and garden centers, 66 in the perimeter of these facilities. Between 2017 and 2021, a total of 127 prohibited plants for sale were seized, 17 positive plants in operator facilities and 10 in the surroundings of a total of 1561 samples.

To carry out the controls in the authorized border control posts (BCP), there is a plant health inspector assigned to the functional area of agriculture and fishing of the Government sub-delegation in Palma. So far, there is no evidence of entries or exits of host plants or specified plants, according to the definition established in Article 1, Decision (EU) 2015/789, and then in Regulation (EU) 2020/1201. In the authorized BCPs, there have been no interceptions of any type of plant material intended for planting as established in the community legislation.

### 2.6. Awareness Campaigns

A main goal of the phytosanitary authorities has been to disseminate information campaigns for main stakeholders and the general public in the detection and control of the pathogen. The list of host plants for Xf is kept updated to help the authorities in charge of carrying out controls on the entry and exit of plant material. About 14 training days have been organized for managers of the national airport management agency (AENA), Port Authority and Ports of the Balearic Islands, and a total of 358 campaigns at points of entry (ports and airports), in which information and dissemination, as well as supply of containers, informative leaflets, and brochures, have been provided. In addition, the Agriculture Service has carried out 379 training campaigns for professional users (garden centers and nurseries) and delivered brochures to publicize the presence of Xf in the Balearic Islands. Moreover, a plant health bulletin board is published monthly, and Xf regular public information is available on the plant health website: Pub-https://www.caib.es/sites/xf (accessed on 15 November 2022) and contact telephone numbers 900 102 186 and an informative email on Xf: sanitatvegetal@dgagric.caib.es

On the other hand, the Forest Health Service has prepared an informative video to show the biology of the bacterium and its vector, as well as the symptoms, signs, and damage they generate, published an edition of the “Visual Guide to symptoms of Xf in forest species of the Balearic Islands”. Periodic updating of information is available on the forest health website: http://sanidadforestal.caib.es. (accessed on 15 November 2022) Consultations about Xf can be done through the contact telephone number (+34 971176666), through the Environmental Information Point (PIA): 900151617 (free), or by email at sanidadforestal@gmail.com. In addition, there have been periodic training on Xf for the Forestry Technical Group and also informative meetings with environmental agents.

## 3. Research on the Reconstruction of the Introduction and Spread of *Xylella fastidiosa* in the Balearic Islands

As mentioned in Article 6 of Decision 2015/789, the Member States should carry out the appropriate investigations to identify the origin of the transferred infected plants. Since 2017, research has been developed to establish the origin and date of the possible entries of the different Xf genotypes on the different islands. From the beginning, there were suspicions that Xf could be behind the great mortality of almond trees that had previously been attributed to the interaction of a complex of wood fungi with drought [37]. In the spring of 2017, it was already known that there were qPCR-positive almond trees scattered throughout the island. MLST analyses carried out in Cordoba, Spain, on leaf samples from Xf-qPCR-positive almond trees pointed out a close genetic relatedness to Xf subsp. *fastidiosa* and *multiplex* that were causing almond leaf scorch disease (ALSD) in California. Even in 2010, the possibility that Xf could be involved in the death of the almond trees had been considered. For all these reasons, in the spring of 2017, we worked on the hypothesis that ALSD could be the cause of the death of almond trees and because the first cases of mass mortality began to be seen in 2007, the bacteria must have been introduced in the early 2000s or earlier.

### 3.1. Disease Incidence

Knowing the disease incidence caused by Xf on different crop diseases was pivotal for several reasons. First, it allowed us to quantify the outbreaks in terms of population rather than scattered infected plant units verified in the laboratory by qPCR. Secondly, it enabled us to properly infer the spatiotemporal spread of the ALSD epidemic, affecting the most important crop in Mallorca. At the same time that the number of samples detected by qPCR was periodically reported as outbreaks notifications to the commission (Figure 1), field studies were carried out to visually estimate the incidence of ALSD throughout the island. A total of 126 almond orchards distributed throughout Mallorca were inspected to estimate the incidence of ALSD in 2017. By counting the trees that showed disease symptoms previously attributed to fungal trunk pathogens as an advanced stage of ALSD, it was visually determined that the incidence of the disease affected approximately 79.5% of the almond trees [38]. To investigate the disease progress, we used the Google Street View panoramic-image repository to approximate the ALSD incidence in 2012. Around 249 orchards distributed throughout the island were visually examined, and the average incidence of ALSD was estimated at 53.4% [38].

On the other hand, there was a certain urgency in knowing the incidence of Pierce’s disease established for the first time in Europe. In the summer of 2018, extensive sampling was carried out in vineyards in Mallorca to verify that the incidence was very heterogeneous. It was found that the disease incidence ranged from less than 1% to 99%, depending mainly on the management of the vegetative cover in spring, the treatments received with insecticides, the age of the plantation, and the varieties planted [39].

Although no specific studies have been carried out to assess the incidence of Xf in wild olive trees, an abundant and widespread species in the Balearic Islands, a very conservative estimate of 10% would indicate hundreds of thousands of infected trees. All these approximations of Xf incidence on crops and forest trees at the end of 2017, transformed into units of infected plants, suggested a range between 1 and 3 million Xf-infected plants in Mallorca alone (*cf*. Figure 1). These differences in the way of describing and interpreting an epidemiological outbreak in a territory deserve further reflection and thus are discussed later in this perspective.

### 3.2. Phylogenetic Analysis

Our second research priority was to establish the geographical origin and the approximate date of introduction of Xf in the Balearic Islands. To do this, we needed to obtain a sufficient number of isolates to sequence their genomes and compare them with other genomes published in GenBank. In an international collaborative study using several Xf isolates of subsp. *multiplex* from Mallorca and Menorca and other isolates collected in Europe and America, it was possible to show that the Balearic isolates were introduced from California [30]. To determine the probable date of introduction, we faced the limitation of the sampling time (2016 to 2019), which was quite scarce to estimate the substitution rate with some confidence and thus calculate the molecular clock. This problem could be solved by anchoring the minimum node date for the Balearic Islands clade to the year of the oldest ring in which Xf DNA sequences were detected in the growth rings of almond trees. Xf DNA sequences of subsp. *fastidiosa* were detected in rings corresponding to 1998 in trees with ALSD symptoms, while Xf. subsp. *multiplex* was found in rings of 2000. This allowed us to properly use the priors in Bayesian inference to estimate with some confidence that the introduction of both subspecies from California occurred around 1990–1997 [38]. By revealing the date and origin of the introduction of the pathogen, another dimension was given to the epidemiological situation of Xf in the Balearic Islands, which cast doubt on the effectiveness of the current phytosanitary measures.

### 3.3. Vector Transmission

To explain how Xf had spread within the islands and to control its transmission, we needed to identify the insect vectors involved, their prevalence, and their ecology. Several research projects were funded by the Balearic Government, the Ministry of Agriculture of the Government of Spain, and the European Food Safety Authority (EFSA) [40]. Early surveys indicated a predominance of *P. spumarius* and much lower proportions of the species *Neophilaenus campestris* in fields of all three islands. In addition, a high density of *P. spumarius* nymphs was observed in different plots between February and April. However, the captures of adults in the vegetative cover and crop canopy were considerably lower. The first detections of infective adults of *P. spumarius* occur in May and accumulate during the summer in variable proportions according to the different studies and the detection methods used [40]. In general, there is a seasonal pattern of the vector populations similar to that observed in Apulia (Italy) in the OQSD [41].

In trials conducted in an insect-proof tunnel, the transmission of Pierce’s disease and ALSD by *P. spumarius* has been demonstrated, after an acquisition access period (APP) and inoculation access period (IAP) of 72–96 h [42]. Cross-transmissions between vineyards and almond trees have also been successfully carried out, and infections of the *multiplex* subspecies of wild olive trees have been transmitted to almond trees. Furthermore, on one occasion, Pierce’s disease was transmitted from an infected to a healthy grapevine through *N. campestris*.

### 3.4. Climatic Conditions

In a recent study carried out in Mallorca, risk maps for Pierce’s disease have been developed for the main wine-producing areas of the world on the basis of epidemiological models [43]. These specific models for the pathosystem *Vitis vinifera*-Xf ST1- *P. spumarius* suggest that the disease could only have become established in Europe on Mediterranean islands, such as the Balearic Islands, or very specific areas in Mediterranean coastlands. The current distribution of Xf in Europe does not seem to be a coincidence, but rather it would indicate where the pathogen could establish itself in a situation where the movement of infected plants in Europe during the last 20 years would have been the norm. These models are important because they reveal that other strains of the pathogen could easily become established in the Balearic Islands or other Mediterranean islands, and they provide an idea of the stochasticity of the invasion process in the initial stage after entry.

### 3.5. Field Observations and Inoculation Experiments

The four factors that intervene in the development of a disease, namely, the hosts, the climate, the insect vectors, and the genetic diversity of the pathogen, have been addressed in these six years of research. On the other hand, studies have been carried out in the field and in insect-proof tunnels on the susceptibility of the different varieties of almond, vine, and olive trees to better understand the diseases and adequately guide the policies aimed at controlling Xf in the Balearic Islands. As a general rule, most almond trees and grapevine varieties are affected by Xf, whereas olive trees are resistant to *fastidiosa* and *multiplex* subspecies. Because grapevines are affected only by subsp. *fastidiosa*, these observations are more conclusive. All grapevine varieties inoculated or monitored in the field are in some way susceptible to Xf. In the inoculation tests, there were significant differences between varieties in the development of symptoms and Xf infection, but part of these differences were due to rootstock–variety interactions. Varietal response to Xf in the inoculations were similar to those observed in the field, although it was not possible to correlate the different incidences with the severities observed in the inoculations due to the intervention of other non-controllable factors such as crop management and its impact on the vector population.

Germplasm bank varietal collections of almond trees naturally exposed to the pathogen have provided quality experimental data to categorize the susceptibility of well-identified crop varieties. The results of the observations in these germplasm banks have been very similar to those in the field. These data have been used to recommend the planting of some tolerant varieties and prohibit the most susceptible ones.

## 4. Discussion

The implementation of phytosanitary measures represents the first combat front for eradicating and containing harmful organisms. If these are taken at the right time and with diligence, they can be very effective in their ultimate purpose. However, this is rarely the case; in particular, pathogenic microorganisms are generally much more difficult to detect than insect pests. In addition, symptoms produced by harmful microorganisms can be non-specific or confused with other root or vascular pathogens, or even with plant physiological disorders. Overall, there is often a significant period elapsing between the introduction event and the detection of the harmful organism, sometimes making posterior eradication or even containment unfeasible. The exposed case of Xf in the Balearic Islands is a good example of this. Between the introduction and its detection, around 23 years passed. Much of the discussion is situated in this context.

In early 2017, the official body soon recognized the infeasibility of the eradication measures established by Decision 2015/789, given the situation of Xf in the Balearic Islands. Technical and political efforts were devoted to convincing the European plant health authorities of the need to adapt the decision for the islands’ specific case. With the entry into force of the modification of the decision and later included in the regulation, the infeasibility of eradication in infected areas is recognized. Instead, it emphasizes the need to intensify surveillance to detect a possible spread of the pathogen. In addition, sampling methods are harmonized in all the delimited areas through the use of the statistical and risk-based sampling tool RIBESS+ developed by the EFSA. Although the benefits of this statistical tool as a surveillance guide are beyond doubt, in areas where Xf has become naturalized, such as the three largest Balearic Islands, the aim of the survey has to be necessarily different to those of detecting, delimiting, and monitoring Xf in buffer zones. Research has shown that there are millions of infected plants in the field and wild borders, so removing only qPCR-positive plants (1360 qPCR+) represents much less than 0.1% of the infected plants. Nowadays, the pathogen and the vector integrate the landscape, so reducing the amount of inoculum below significant levels would imply the elimination of millions of plants and the massive use of insecticides. In support of this argument, a recent model for the ALSD epidemic in Mallorca has estimated that the mean value of the basic reproductive number (*R_0_*) < 1 occurred approximately in 2011. This indicates that the number of new infections has been decreasing year after year and the trend is directed to the extinction of the epidemic if there is no replacement of susceptible plants [43]. All this suggests that instead of directing the effort to indiscriminately detect new plants infected with qPCR, the focus should be on reducing the inoculum in new plantations and in areas where the incidence is still low, such as in the mountains or endangered ecosystems far away from crops.

Except in mountain areas, the extreme southwest of the island of Mallorca, the north of Menorca, and the little island of Formentera, Xf is not spreading in the sense of increasing an epidemic front. As explained above, the pathogen has been established for decades on the islands and is everywhere there are susceptible hosts. In some crops such as the almond tree, the spread on a regional scale is even decreasing, since the incidence is very high, and few trees remain uninfected. The idea of monitoring Xf spread makes little sense in the case of the Balearic Islands. It is not that the usefulness of the RIBESS+ statistical tool is questioned, but we believe that the aim of the surveys has to be redefined to target other risks, having in mind that the pathogen has been around for a long time. Our suggestion would be to focus on selective surveys aimed at investigating specific risks, such as monitoring the introduction of new strains of Xf or the introduction of strains between islands as well as new potential insect potential vectors, which seems more reasonable.

This connects with another widespread but probably misconceived idea about the emergence of new genotypes. Although much remains to be learned, homologous recombination in Xf appears to be a major driver in host shifts [17]. Jumps to new hosts could be more abrupt than gradual, that is, the pathogenicity and virulence of the genotypes would be established from the first contact with the new host and not progressively due to the accumulation of point mutations in the pathogen’s genome. For example, we assume that the ST81 of subsp. *multiplex* must have been virulent from the first contact with wild olive trees in Mallorca, but instead, it rarely infects nearby olive trees. We do not expect, though we cannot discard, that Xf colonies living as commensals on olive trees will progressively become more aggressive through the accumulating mutations or incorporation of genetic material from other species. In other words, new host–Xf strain contacts would be under extreme episodic selection. On the contrary, in the case of ALSD and Pierce’s disease in Mallorca, these strains and diseases have been transferred from California to Mallorca. They are not new contacts and, therefore, have been under stabilizing selection for some time [44]. After 27 years since their introduction, it is not expected that there will be a sudden outbreak in a new host without recombination events, the introduction of a new pathogen strain, or the intervention of extreme environmental factors.

We believe that when Xf becomes naturalized, as has occurred in Corsica or the Balearic Islands, the best strategy to detect new variants of the pathogen is to focus on vectors, as has been recently done in Corsica [45]. The main reason is that if genetic recombination occurs, the fittest new genotype will accumulate in the newly formed soft tissue of the infected plant where vectors feed more frequently. Colony bacterial adhesion to the walls of insect mouthparts would act as a sink where foreign DNA could be detected through adequate molecular methods. Therefore, the likelihood of detecting the new variant would be expected to be greater in the vector mouthparts than in a tree with hundreds of twigs. The other reason that needs more theoretical and experimental support is that genetic recombination between Xf strains might occur more frequently in vectors than in hosts.

Varietal resistance/tolerance to Xf has been identified as a strategy to control the pathogen in crops [46]. It is believed that greater genetic diversity in the host population protects against possible epidemic outbreaks; however, this well-established paradigm was not fulfilled in the case of ALSD in the Balearic Islands where both fastidiosa and multiplex subspecies spread within and among almond orchards (>100 almond tree varieties) of the island of Mallorca for approximately three decades. On a regional scale, the decision to at least prohibit new plantations of the most susceptible varieties in order to reduce the inoculum within new plantations seems accurate, as has been done in the Balearic Islands. In Majorcan viticulture, the average size of the vineyard ranges from 1 to 10 hectares. Rows of different varieties are usually found in the same plot (Figure 4), so monoculture of more tolerant varieties to the bacteria does not seem to be a viable or realistic strategy for controlling Pierce’s disease, given the large number of wineries in a small territory. Instead, the use of more tolerant almond varieties would make more sense as it is a more industrial crop than vines and does not depend as much on organoleptic qualities.

## 5. Future Perspectives

There is no choice but to live with Xf in the Balearic Islands. Knowing how to live together, however, should not only become a positivist slogan but should also recognize that there are dangerous diseases established whose control can be achieved at a reasonable cost. Simple actions, such as the control of nursery material, weed control, and the elimination of infected plants in new vineyards or almond plantations make cultivation of main crops possible with little economic loss.

It is also important to anticipate new possible risks. The climate of the Balearic Islands and other Mediterranean islands seems to be suitable for all Xf subspecies [47]. Therefore, we believe that efforts should focus on preventing new entries of the pathogen or potential vectors through international plant trade. Finally, this review aims to share the experience of Xf control measures in the Balearic Islands with other areas where the pathogen has not yet been introduced or detected. We have exposed the facts and the action plans applied for the control of bacterium and vectors in the Balearic Islands, as well as the advances in the knowledge of the origin, phylogenetic relationship, and dating of the introduction of the two Xf strains in Majorca. We believe that the experience of Xf in the Balearic Islands invites us to consider a more epidemiological approach in the legislation of harmful organisms, in which it contemplates the possibility of unexpected events that force the adoption of control measures proportional to the real epidemiological situation of that territory.

## Figures and Tables

**Figure 1 microorganisms-10-02393-f001:**
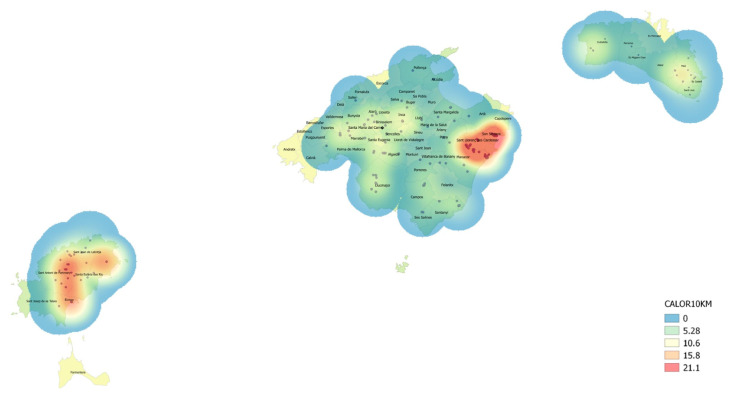
Density map of the number of Xf-qPCR+ samples per Km at the first Audit DG SANTE in June 2017. Demarcated 10 Km areas in blue circles.

**Figure 2 microorganisms-10-02393-f002:**
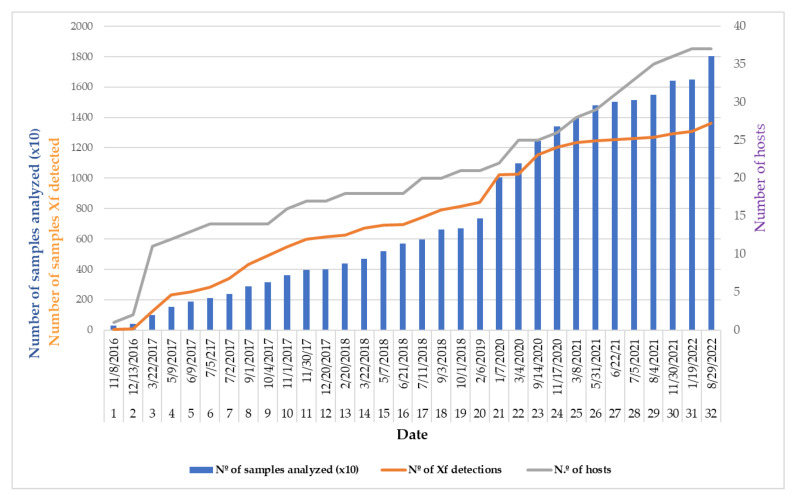
Progression of the number of total samples, Xf-qPCR-positive samples, and hosts over time in the Balearic Islands.

**Figure 3 microorganisms-10-02393-f003:**
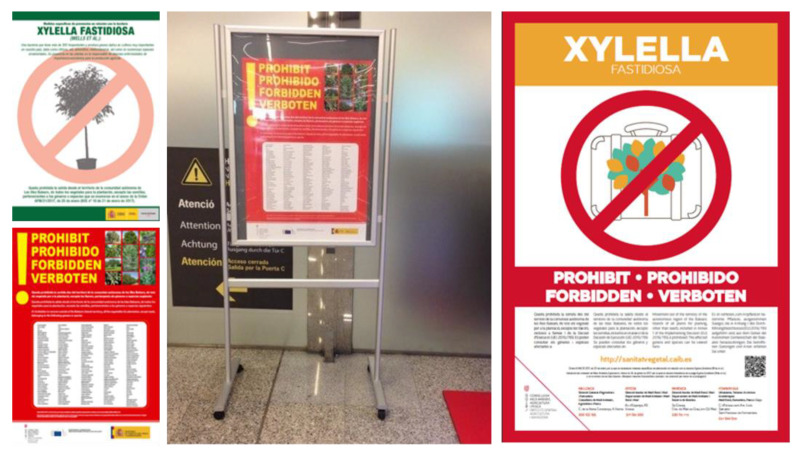
Informative, leaflet brochures, and posters of *Xylella fastidiosa* at the airport of Palma de Mallorca.

**Figure 4 microorganisms-10-02393-f004:**
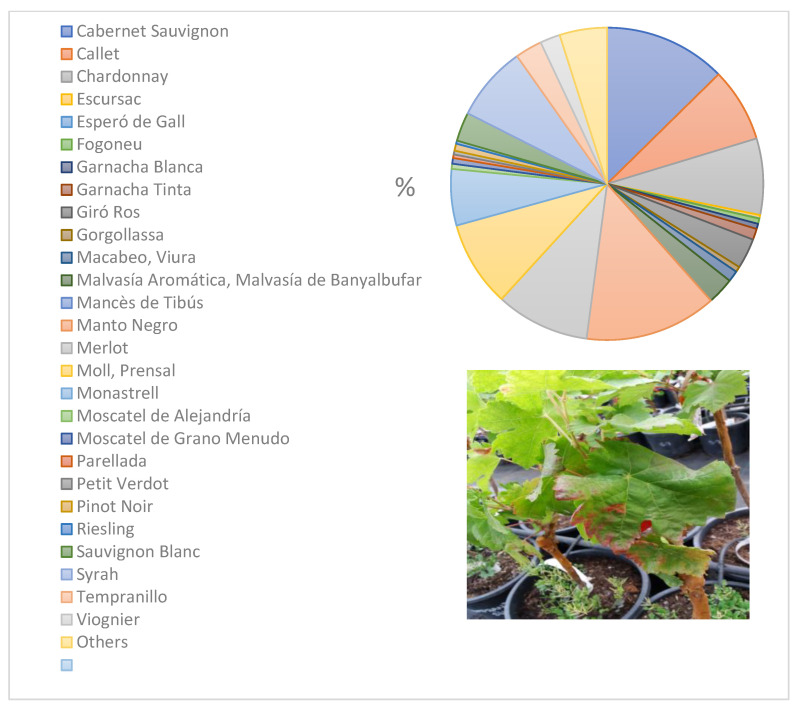
Diversity and surface-area proportions of grapevine varieties planted in Mallorca. All these varieties have been shown to be susceptible *to Xylella fastidiosa* in the field and in inoculation experiments. Symptoms of a grapevine (var. Tempranillo) seven weeks after inoculation (bottom).

**Table 1 microorganisms-10-02393-t001:** Host plants for *Xylella fastidiosa* recorded in the Balearic Islands in 2022.

Host	Family	Islands ^1^	Strain ^2^
*Acacia saligna*	Leguminosae	Ma|Ib	ST81|ST80
*Calicotome spinosa* *Cistus albidus* *Cistus monspeliensis* *Clematis cirrhosa*	Leguminosae Cistaceae Cistaceae Ranunculaceae	Ma Ma|Me|Ib Ma Me	ST1 ST81|ST81|ST80 ST1 ST81
*Elaeagnus angustifolia*	Elaeagnaceae	Ib	Not determined
*Ficus carica*	Moraceae	Ma|Me	ST81|ST81
*Fraxinus angustifolia* *Genista hirsuta* *Genista lucida* *Genista valdes-bermejoi* *Helichrysum stoechas* *Juglans regia* *Lavandula angustifolia* *Lavandula dentata* *Nerium oleander* *Olea europaea var. europaea* *Olea europaea var. sylvestris* *Phagnalon saxatile* *Phillyrea angustifolia* *Phlomis italica* *Polygala myrtifolia* *Prunus avium* *Prunus domestica* *Prunus dulcis* *Rhamnus alaternus* *Rosmarinus officinalis* *Ruta chalepensis* *Santolina chamaecyparissus* *Santolina magonica* *Salvia officinalis* *Spartium junceum* *Teucrium capitatum* *Thymus vulgaris* *Ulex parviflorus* *Vitex agnus-castus* *Vitis vinifera*	Oleaceae Leguminosae Leguminosae Leguminosae Compositae Juglandaceae Labiatae Labiatae Apocynaceae Oleaceae Oleaceae Compositae Oleaceae Labiatae Polygalaceae Rosaceae Rosaceae Rosaceae Rhamnaceae Labiatae Rutaceae Compositae Compositae Labiatae Leguminosae Labiatae Labiatae Leguminosae Verbenaceae Vitaceae	Ma Ib Ma Ma Ma|Me Ma Ma|Ib Ma|Ib Ma|Ib Ma|Me|Ib Ma|Me|Ib Ma Ma Me Ma|Ib Ma Ma Ma|Me|Ib Ma|Me Ma|Me|Ib Ma Ma|Me Me Ma Ma Ma Ib Ib Me Ma	ST81 Not determined ST1 ST81 Not determined|ST81 ST1 ST81|ST80 ST81|ST80 ST81|Not determined ST81|ST81|ST80 ST81|ST81|ST80 Not determined ST81 Not determined ST1, ST7, ST81|ST80 ST1 ST81 ST1, ST7, ST81|ST81|ST80 ST1, ST81|ST81 ST81|ST81|ST80 ST1 Not determined|ST81 ST81 ST81 ST81 ST1 Not determined ST80 Not determined ST1

^1^ Ma= Mallorca; Ib= Ibiza; Me= Menorca. ^2^ ST= sequence types found in the Balearic Islands.

## Data Availability

Not applicable.

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
