# Peer review of "Evaluation of Control Strategies for Xylella fastidiosa in the Balearic Islands"

_microorganisms, 2022, doi:10.3390/microorganisms10122393_

Round 1

Reviewer 1 Report

The manuscript by Quetglas et al. provides a thorough historic assessment regarding dissemination of the phytopathogenic bacterium Xylella fastidiosa in the Balearic Islands, along with the  main strategies employed by the Spanish government to control this economically important plant pathogen. Overall, the manuscript is sound and adequate to be published in a special issue entitled “Plant Pathogenic Microorganisms: State-of-the-Art Research in Spain".

Nevertheless, before it can be accepted for publication, I recommend that the manuscript be thoroughly revised by a native English speaker, since there are several minor mistakes throughout the text that compromise clarity (too many to be corrected in a peer-review process). More important, authors MUST provide new versions of Figures 1, 3 (both lack definition) and 4 (labels in the pie chart cannot be properly read/identified).

Author Response

We apologize for the editing of English Language.

The manuscript has been now carefully revised by the corresponding author and a native English speaker and are marked using the Track Changes function in Word.

We hope that the extensive editing of English language and style carried out would improve the clarity of the manuscript.

We include new versions of figures 1,3 and 4 with higher definitions.   

Reviewer 2 Report

The article submitted by Quetglass et al. is a detailed description of the phytosanitary measures applied in the Balearic Islands to face Xylella fastidiosa.

The paper describes the surveys, detection, host plants infected by the bacterium and subsequent elimination, vector activities as well as some molecular features found in the Xylella fastidiosa strains isolated in the Balearic Islands. A connection is also supposed with drought.

The paper is quite informative and provides an useful base to frame the situation in the area.

However, since the Authors offer a view on control strategies, they should include into the Introduction some links to the current efforts that are going on in the U.S.A. and Italy to control the pathogen in the field through some formulates that show positive results to reduce the level of tha pathogen concentration within the tree.

Author Response

We appreciate the Reviewer comments.

We agree with the need to include some information on the efforst to control the pathogen in Italy and the USA.

We now have included the following sentence:

“Several minerals and compounds in addition to microbial endophytes as biocontrol agents have been tested in America and Italy showing some protection against Xf infections in grapevines [14], citrus [15] and olive trees[16]; however, their large-scale application is still economically expensive [13].”

At the end of the introduction, we remark that there are recently published reviews of general control measures against the pathogen:

Our purpose is to focus exclusively to control strategies in the Balearic Islands as excellent general reviews of control attempts and management of Xf have been recently published [30]”

Regards

Eduardo